# Multi-Effect Enhanced Raman Scattering Based on Au/ZnO Nanorods Structures

**DOI:** 10.3390/nano12213785

**Published:** 2022-10-27

**Authors:** Yi Lin, Jun Zhang, Yalan Zhang, Sai Yan, Feng Nan, Yanlong Yu

**Affiliations:** Faculty of Mathematics and Physics, Huaiyin Institute of Technology, Huai’an 223003, China

**Keywords:** surface-enhanced Raman scattering, nano-composite structure, Au nanoparticles, ZnO nanorod arrays, localized surface plasmon resonance, whispering gallery mode resonance

## Abstract

Surface-enhanced Raman scattering (SERS) was considered a potential spectroscopic technique for applications of molecular detection and has drawn great research interest during the past decade. So far, fabrications of cost-effective SERS substrates with high sensitivity and stability and the corresponding enhanced mechanisms are always among the list of research topics, although great progress has been made. In this work, Au particles were decorated on Si, ZnO film and ZnO nanorod arrays simultaneously by an economical method of ion sputtering, generating three kinds of SERS substrates for R6G detection. The morphology difference of Au particles on different samples and the consequent influence on Raman scattering were studied. The experiment results exhibited that substrates with Au particles decorated on ZnO nanorods had the highest Raman enhancement factor. Furthermore, multi-effect enhanced mechanisms summarized as localized surface plasmon resonance (LSPR) filed coupling, electron transferring induced by LSPR of Au particles and whispering gallery mode (WGM) effect of the ZnO cavity were presented. This work provides a convenient and efficient method of fabricating SERS substrates and indicates that such proper metal/semiconductor composite structures are promising candidates for SERS applications.

## 1. Introduction

Raman scattering is a typical inelastic scattering behavior between photons and other particles. The energy difference between incident and scattered photons depends not on the incident light frequency but on the characteristic vibrational properties of target molecules. As a vibrational spectroscopic technique, Raman scattering has the advantages of quick response, simplicity in the pretreatments of samples, and, most of all, high ability to recognize specific molecules. However, its disadvantage of low sensitivity due to a small scattering cross-section prevents it from being widely used in practical applications [1]. In 1974, a remarkable enhancement of the Raman signal from pyridine attached to a rough silver electrode was first discovered by Fleischmann M. [2]. Later, Jeanmaire D.L. and Kneipp K. observed more similar phenomena and named this effect surface-enhanced Raman scattering (SERS) [3,4].

With the development of nanofabrication techniques, Nie S.M. firstly achieved the Raman signal detection at the single-molecule level by attaching rhodamine 6G (R6G) molecules to silver colloidal nanoparticles as early as 1997 [5] and many other significant signs of progress in this area have also been made during the last decades [6,7,8,9,10,11,12,13]. One of the goals of SERS research is to actualize its potential applications as an ultrasensitive detection technique. The primary difficulty in achieving this goal is the preparation of a stable and cost-effective SERS substrate with a high enhancement factor. In the early stages, great efforts have been made to synthesize different metal colloidal nanoparticles, typically Ag and Au, as SERS substrates. By controlling the sizes, shapes and compositions of colloidal particles, a giant enhancement factor of up to 14 orders has been reached [14,15,16,17]. Such a strong enhanced effect required the aggregation of isolate particles, forming the colloidal clusters, which could be served as ‘hot spots’ during the scattering process. The enhancement factor depended on the morphology properties and spatial distributions of these clusters, while both of which were somehow uncontrollable. To overcome these defects, many researchers tried to design SERS substrates with innovative structures, such as periodic arrays of nanoparticles, core-shell composites based on a flexible substrate and highly ordered metal nanorods [18,19,20,21,22,23]. However, the preparations of such SERS substrates were uneconomic and time-consuming to some degree because of using particular methods, such as nanosphere lithography, electron beam lithography and focused ion beam milling. In recent years, researchers found that some kinds of semiconductors can also be SERS active, providing an enhancement factor of 10^1^ to 10^3^ orders [24,25,26,27,28]. Among various semiconductors, ZnO has drawn a lot of attention due to its advantages of simplicity in preparation, diversity in structures, good chemical stability and high bio-compatibility. Hence, different kinds of metal-ZnO hybrid structures have been designed and synthesized for versatile applications, such as photocatalysis [29], molecular detection and biosensing [30,31,32,33].

Although inspiring results about SERS substrates consisting of different metal/semiconductor structures have been reported, some important issues still deserve further studies. For example, the combination of nanoscale metal particles and semiconductors will cause interactions between them, leading to some particular physical processes, including electron transferring and band change. Such processes will be positively motivated by light irradiation if the photon energy of incident light is coupled with the LSPR resonance of metal particles. Consequently, this may have an extra effect on the enhancement of Raman scattering, while the underlying mechanism still needs to be studied. Besides, for the composite SERS substrates, the total enhancement effect may depend on the morphologies and structures of the semiconductors, but relevant comparative studies have rarely been reported at present. Herein, in this work, using a low-cost ion sputtering device, we fabricated three kinds of SERS substrates by decorating Au nanoparticles on Si, ZnO films and ZnO nanorods. Morphology characterizations demonstrated that Au particles with good uniformity were decorated on all substrates. Then, eight samples with a different sputtering time for each kind of substrate were used to detect the Raman signal of R6G at exactly the same conditions. The experiment results showed that all three kinds of substrates exhibited an obvious Raman-enhanced effect, while the one based on ZnO nanorods had the highest enhancement factor. Furthermore, the slight distinctions of Au particles on different samples and the consequent influence on Raman scattering were discussed. At last, a multi-effect enhanced mechanism summarized as LSPR filed coupling, electron transferring induced by LSPR of Au particles and WGM effect of ZnO cavity was presented.

## 2. Materials and Methods

### 2.1. Preparation of ZnO Films

ZnO film was prepared on Si substrates by the method of magnetron sputtering. First, two carefully washed Si substrates with sizes 10 mm × 10 mm were fixed on the center of the sample stage and were aimed by the ZnO target before the reactor chamber was vacuumed to 5 × 10^−4^ pa. After that, Ar and O_2_ with a mass flow ratio of 50:5 were introduced as reaction gases, and the chamber pressure was precisely set to 2 Pa. After 30 min deposition at 100 W sputtering power, two ZnO film substrates were taken out and evenly cut into 8 pieces for further treatment.

### 2.2. Preparation of ZnO Nanorod Arrays

ZnO nanorod arrays (NRs) were prepared by the hydrothermal method. Firstly, ZnO seeds were deposited on two Si substrates with sizes 10 mm × 10 mm by magnetron sputtering for 10 s under the same sputtering conditions mentioned above. Secondly, the hydrothermal experiment was conducted. Specifically, 0.439 g of zinc acetate and 0.28 g of methenamine were dissolved in 100 mL of deionized water to formulate a mixture solution with an equal molar concentration of 25 mM. Then the well-stirred solution was transferred into a Teflon container of a stainless steel autoclave, and the as-prepared Si substrates were immersed in it with the seed layer side down. After that, the sealed autoclave was placed in an electric drying oven for 3 h for a hydrothermal reaction at 90 °C. Finally, after the reaction finished, two substrates were washed and dried and then cut into 8 pieces.

### 2.3. Decoration of Au Nanoparticles on As-Prepared Substrates

Three kinds of SERS substrates, named as Au@Si, Au@ZnO-film and Au@ZnO-NRs, were fabricated simultaneously by a desktop ion sputter. Technically, three different substrates and a transparent quartz substrate which was used to measure the optical properties of Au nanoparticles were put together as one set in the center of the sample stage. Then the chamber was sealed and pumped to 1 Pa before Ar was introduced as working gas. After the chamber was washed three times by Ar flow, the operating pressure and current were adjusted to 30 Pa and 10 mA to carry out the sputtering process. Under these conditions, eight sets of samples were fabricated, with the sputtering time increased from 20 s for the first set to 160 s for the last one. The fabrication process of one set is demonstrated in Figure 1a.

### 2.4. Characterization

The morphology of SERS substrates was characterized by scanning electron microscopy (Carl Zeiss, Oberkochen, Germany, Ultra Plus). Raman signals of R6G solution with a concentration of 5 × 10^−8^ M were detected by a Laser confocal Raman micro-spectrometer (HORIBA, Kyoto, Japan, LabRAM HR 800). Specifically, 10 μL R6G ethanol solution was dropped on each sample, then a 514 nm laser beam was introduced for excitation and was carefully focused on the sample to ensure the data accuracy. Absorption spectra of Au nanoparticles on quartz with different sputtering times were detected by an ultraviolet-visible spectrophotometer (Shimadzu, Kyoto, Japan, UV-3600).

## 3. Results and Discussion

### 3.1. Structure and Morphology

The SEM images of as-prepared ZnO film and ZnO nanorods are shown in Figure 1b,c. It can be clearly seen from Figure 1b that ZnO film was actually composed of continuous crystal grains with diameters of about 30 nm and had a pure and clean surface before Au sputtering. Figure 1c shows that ZnO nanorods with diameters in the range of 350 to 400 nm were uniformly and vertically grown on a Si substrate. Before Au sputtering, each nanorod had almost perfect smooth surfaces. The SEM images of Au@Si, Au@ZnO-film and Au@ZnO-NRs substrates taken under the same magnification were shown in Figure 2, Figure 3 and Figure 4, respectively. As shown in Figure 2a, when the sputtering time was 20 s, Au particles with extremely small sizes appeared indistinctly on the substrate. As sputtering time increased, approximately circular particles were formed, and both their sizes and density gradually got larger. When the sputtering time reached 80 s, as shown in the red circles of Figure 2d, some irregular island structures emerged. The proportion and size of these island structures got larger as the sputtering time further increased. For the substrate with a sputtering time of 160 s, the irregular island structures became dominant, and their sizes were in the range of 20 to 30 nm. From Figure 3a–g, it can be seen that Au particles had not been apparently formed on ZnO film when the sputtering time was less than 40 s. Compared to Au@Si substrates, for the same sputtering time, the particle sizes of Au@ZnO-film were a little smaller and more uniform. When the sputtering time was 160 s, as shown in Figure 3h, all the ZnO grain surfaces were fully coated by Au particles whose sizes were in the range of 15 nm to 25 nm. Figure 4 showed that a single ZnO nanorod exhibited a typical hexagonal structure, and Au particles started to be clearly visible on both the top and side faces of it when the sputtering time reached 60 s. For the sample with a sputtering time of 160 s, as shown in Figure 4h, the entire surface of the ZnO nanorod was covered by Au particles with a size in the range of 10 nm to 15 nm. Compared to Au@Si and Au@ZnO-film substrates, for the same sputtering time, Au particles of Au@ZnO-NRs had the smallest sizes and the most uniform distributions. In general, Au particles with good uniformity had been fabricated on all three kinds of substrates by the adopted method, exhibiting some subtle distinctions in particle sizes, particle densities and gap sizes.

### 3.2. Raman Detection of R6G

The Raman signals of R6G detected from different substrates were displayed in Figure 5. For Au@Si substrates, as shown in Figure 5a, when the sputtering time was less than 120 s, only a fluorescence background appeared, and no notable Raman peaks were found in the spectra. As the sputtering time reached 120 s, four characteristic Raman peaks at 612, 1363, 1509 and 1650 cm^−1^ were detected, and four new peaks at 773, 1184, 1311 and 1574 cm^−1^ appeared as sputtering time further increased. Compared to Au@Si substrates, Au@ZnO-film and Au@ZnO-NRs substrates were much more sensitive. As can be seen from Figure 5b,c, the first four Raman peaks had already been detected from the Au@ZnO-film substrate with a sputtering time of 60 s and from the Au@ZnO-NRs substrate with a sputtering time of 40 s. After Raman signals were detected, the peak intensities were gradually enhanced as sputtering time increased. Roughly speaking, Au@ZnO-NRs had the best-enhanced effect among the three kinds of substrates.

To study the difference of Raman enhanced effect in detail, Raman spectra detected from three kinds of substrates with the same sputtering time were compared in Figure 6. It can be seen that, for each set of samples, Au@ZnO-NRs had the largest peak intensity, while Au@Si had the smallest one. The enhancement factor (*EF*) is an essential indicator of the sensitivity of the SERS substrate, which can be calculated by the formula expressed as
(1)$$EF=\frac{I_{SERS}∕N_{SERS}}{I_{Normal}∕N_{Normal}}$$

In this formula, *I_SERS_* and *I_Normal_* are the Raman scatting intensities detected from the SERS-active substrate and normal substrate, which is not decorated by noble metal particles. *N_SERS_* and *N_Normal_* are the numbers of molecules involved in the detection, which usually are represented by the molecular concentrations. In this work, pure Si was employed as the normal substrate, and in order to obtain the Raman signals, an R6G solution with a high concentration of 0.1 M was dropped on it. To estimate the EF of different SERS substrates, Raman peaks at 612 cm^−1^ detected from three kinds of samples with the same sputtering time of 160 s were selected for comparison. As shown in Figure 7, the relative intensities of Au@ZnO-NRs, Au@ZnO-film and Au@Si were 4725, 2106 and 1012, respectively, while the value of pure Si substrate was just 613. By computation, the enhancement factors of Au@ZnO-NRs, Au@ZnO-film and Au@Si were 1.54 × 10^7^, 7 × 10^6^ and 3.36 × 10^6^.

### 3.3. Discussion of Enhanced Mechanism

So far, surface-enhanced Raman scattering is mostly attributed to the localized surface plasmon resonance (LSPR) of noble metal nanoparticles. Normally, Raman scattering intensity *I_RS_* can be expressed as $I_{RS}=N\sigma I_{Laser}$, where *N* is the number of molecules involved in scattering, σ the scattering cross section and *I_LASER_* the excitation laser intensity. When molecules are adsorbed on the metal particles, the formula transforms to $I_{SERS}=N\sigma A_{S}^{2}A_{L}^{2}I_{Laser}$, in which *A_S_* and *A_L_* represent enhanced coefficients of scattering light field and excitation light field. When resonant coupling occurs between incident photons and localized surface plasmons of metal particles, the localized field can be greatly enhanced, causing the significant effect of SERS [1].

For further study of the enhanced mechanism, absorption spectra of Au nanoparticles on quartz substrates with different sputtering times were measured. As shown in Figure 8a, the absorption peaks of Au particles gradually shifted from 525 nm to 554 nm as sputtering time increased from 20 s to 160 s. Figure 8b shows the corresponding SEM images of Au nanoparticles on quartz. Compared to Figure 2, Figure 3 and Figure 4, it can be seen that Au particles on these four kinds of substrates had basically the same change law of sizes and shapes. As is known, the absorption of Au particles is attributed to their LSPR resonances which mostly depend on their sizes and shapes. So, these four kinds of substrates should share approximately the same absorption properties. It can also be seen from Figure 8a that for each case, the excitation wavelength of 514 nm was included in the broad absorption spectrum, which means the resonant coupling condition was always fulfilled. Consequently, both the scattering light field and excitation light field were enhanced, causing Raman scattering enhancement. With the increment of sputtering time, the particle sizes and densities got larger and the distances between particles got smaller, generating SERS ‘hot spots’, which significantly increase the enhanced effect [34]. Especially for Au@ZnO-NRs substrate with 160 s sputtering time, the distance between particles was less than 5 nm, which would cause the field coupling of LSPR of adjacent Au particles, and lead to giant enhanced Raman scattering activity.

Besides the field coupling, there exists another enhanced mechanism that relies on the transferring of excited electrons [35,36,37]. Under proper excitation, the resonant surface electrons of Au particles can be transferred to R6G molecules, promoting the molecular vibration, which in turn increased the scattering cross-section, and consequently, Raman scattering effect was enhanced. For substrates of Au@ZnO-film and Au@ZnO-NRs, ZnO could provide an extra approach for electron transferring from Au particles to R6G molecules, so these two kinds of substrates exhibit better SERS effect than Au@Si. Finally, for Au@ZnO-NRs, the WGM effect in the hexagonal ZnO cavity could further improve the Raman sensitivity. When the 514 nm excitation light was incident on Au@ZnO-NRs substrates, parts of light photons interacted directly with R6G molecules and Au nanoparticles, generating primary Raman scattering. Some other parts of photons went into the ZnO cavity, forming WGM resonant light due to the total reflection on the hexagonal ZnO/air boundary [38]. When the WGM light had the chance to emit from the ZnO cavity, it interacted with R6G molecules and Au nanoparticles again, generating secondary Raman scattering and leading to a higher Raman enhancement factor. The multi-effect enhanced mechanism of Au@ZnO-NRs substrates is illustrated in Figure 9.

## 4. Conclusions

In summary, ion sputtering, a convenient and cost-effective method, was employed to decorate Au nanoparticles on the surface of Si, ZnO film and ZnO nanorods, preparing three kinds of SERS substrates for R6G detection. Morphology characterizations showed that Au particles with uniform sizes and distributions were successfully fabricated on all samples. Comparison measurements of Raman detection indicated that Au@ZnO-NRs substrates exhibited the highest SERS sensitivity, which was attributed to a multi-effect enhanced mechanism. First, the LSPR of Au particles could directly enhance the local field of both incident light and scattering light. Second, electron transfer induced by the particular metal-semiconductor composite structure could promote molecular vibration, which would increase the scattering probability. Third, the WGM effect of incident light in the ZnO cavity would produce more opportunities for light-molecule interactions, leading to more scattering behavior. Two other advantages of such composite structures are also concluded as follows. First, the three-dimensional nanostructure of the as-prepared substrate is beneficial for adsorbing more molecules. Second, good biocompatibility of ZnO makes this substrate suitable for bio-modification and hence for biological detection. Therefore, our study shows that Au/ZnO nanorods composite structure is an appropriate SERS substrate and has great potential for practical applications.

## Figures and Tables

**Figure 1 nanomaterials-12-03785-f001:**
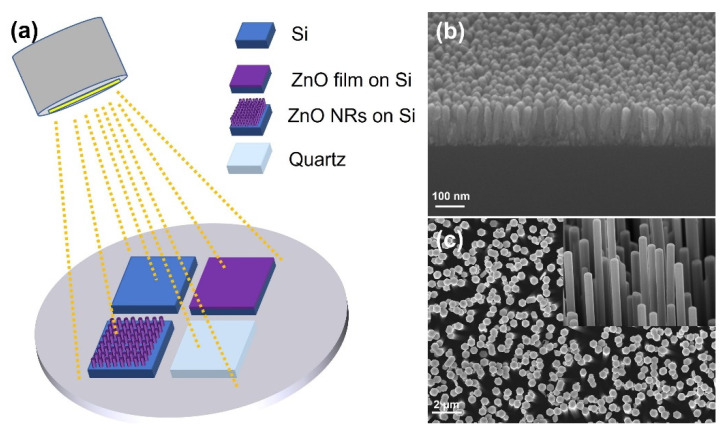
(**a**) Schematic illustration of Au sputtering on four kinds of substrates; (**b**) Tilted cross-section SEM image of ZnO film before Au sputtering; (**c**) Top-view SEM image of ZnO nanorods arrays before Au sputtering; the inset is the corresponding side-view SEM image of individual ZnO nanorods.

**Figure 2 nanomaterials-12-03785-f002:**
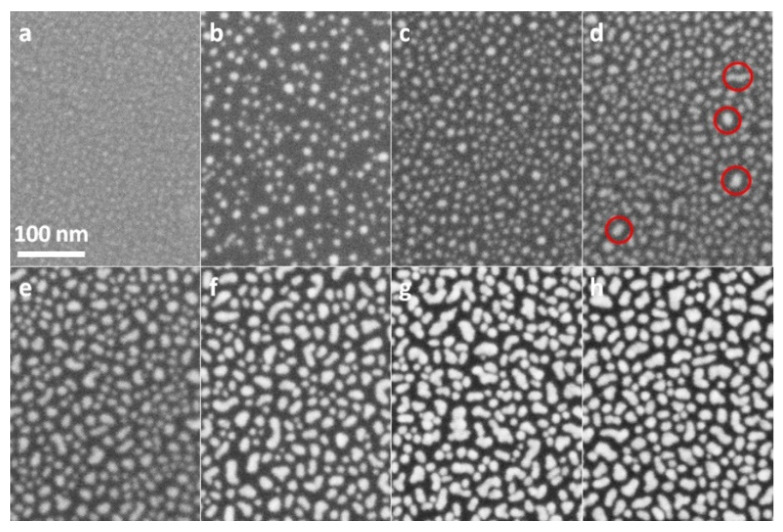
SEM images of Au@Si substrates with sputtering time of 20 s (**a**), 40 s (**b**), 60 s (**c**), 80 s (**d**), 100 s (**e**), 120 s (**f**), 140 s (**g**), and 160 s (**h**).

**Figure 3 nanomaterials-12-03785-f003:**
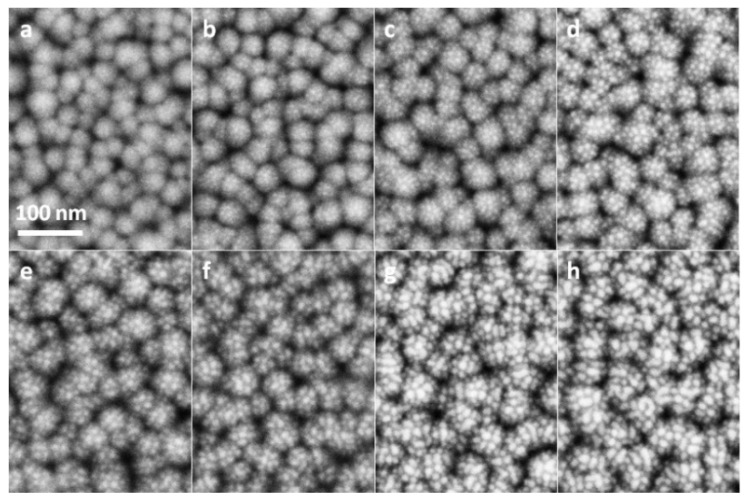
SEM images of Au@ZnO-film substrates with sputtering time of 20 s (**a**), 40 s (**b**), 60 s (**c**), 80 s (**d**), 100 s (**e**), 120 s (**f**), 140 s (**g**), and 160 s (**h**).

**Figure 4 nanomaterials-12-03785-f004:**
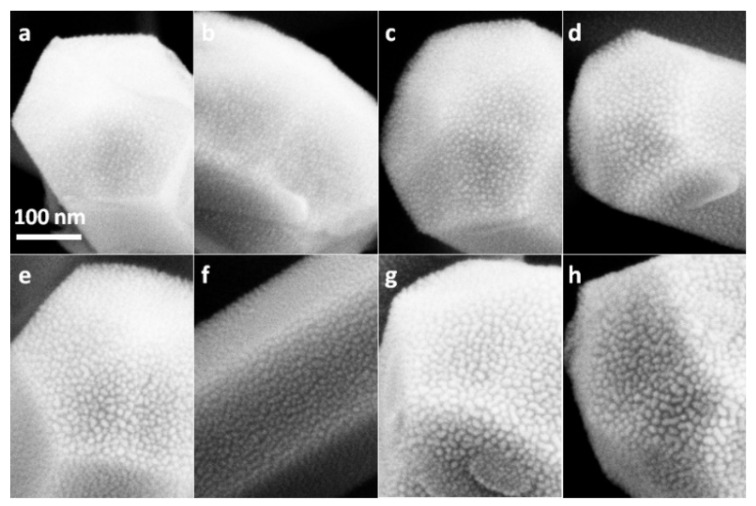
SEM images of Au@ZnO-NRs substrates with sputtering time of 20 s (**a**), 40 s (**b**), 60 s (**c**), 80 s (**d**), 100 s (**e**), 120 s (**f**), 140 s (**g**), and 160 s (**h**).

**Figure 5 nanomaterials-12-03785-f005:**
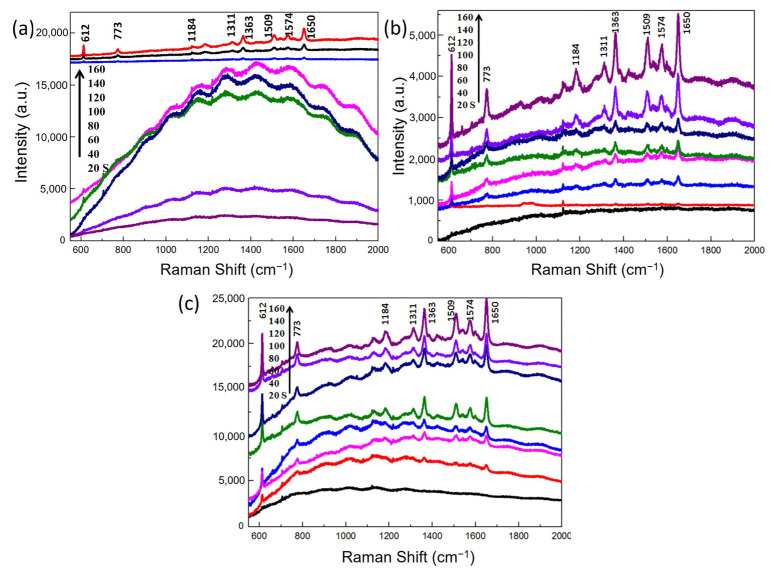
Raman spectra of R6G detected from three sets of SERS substrates, Au@Si (**a**), Au@ZnO-film (**b**) and Au@ZnO-NRs (**c**).

**Figure 6 nanomaterials-12-03785-f006:**
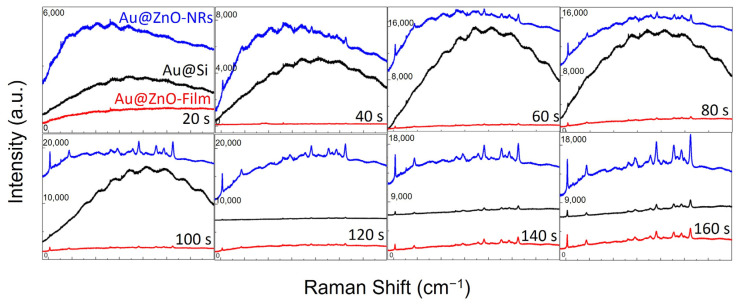
Raman spectra detected from three kinds of substrates with the same sputtering time.

**Figure 7 nanomaterials-12-03785-f007:**
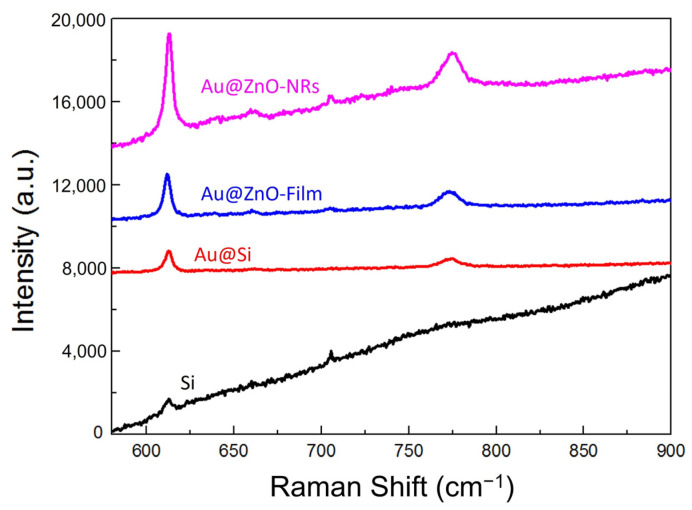
Raman spectra at 612 cm^−1^ detected from three kinds of substrates with a sputtering time of 160 s, and from Si substrate.

**Figure 8 nanomaterials-12-03785-f008:**
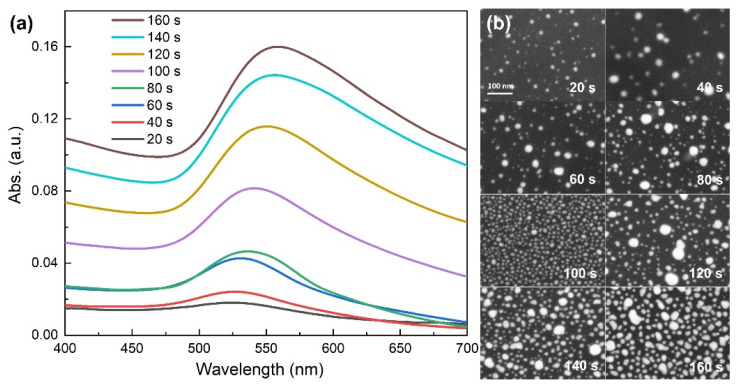
(**a**) Absorption spectra of Au nanoparticles on quartz substrates with different sputtering times; (**b**) The corresponding SEM images of Au nanoparticles.

**Figure 9 nanomaterials-12-03785-f009:**
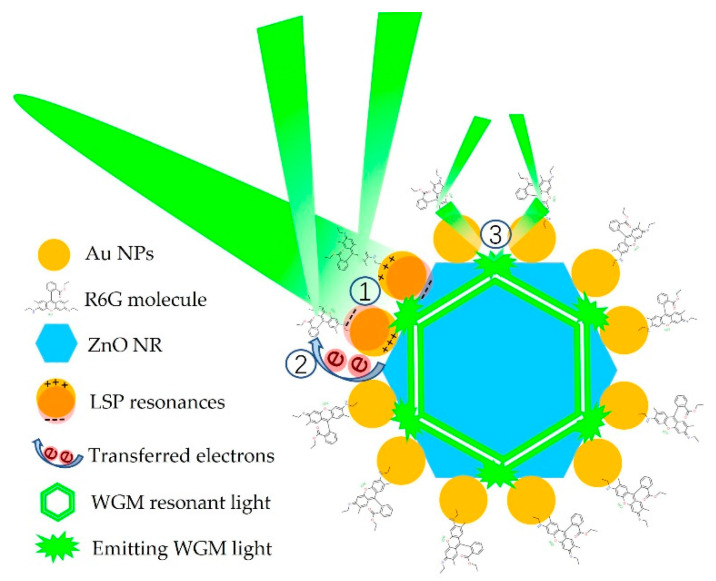
Schematic diagram of multi-effect Raman enhanced mechanism of Au@ZnO-NRs: ① LSPRs of Au NPs enhance the light field; ② Transferred electrons from Au to R6G increase the scattering cross section; ③ WGM resonant light emitting from ZnO cavity provides secondary Raman scattering.

## Data Availability

All data presented in this study are available on request from the corresponding author.

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
