# Peer review of "Multi-Effect Enhanced Raman Scattering Based on Au/ZnO Nanorods Structures"

_nanomaterials, 2022, doi:10.3390/nano12213785_

Round 1

Reviewer 1 Report

The paper represents an interesting study using surface enhanced Raman spectroscopy on substrates of three kinds: Si wafer, ZnO film and ZnO nanorod arrays where Au particles were deposited using an ion beam method. Interestingly , the experiment showed that substrates with Au particles decorated on ZnO nanorods had the highest Raman enhancement factor. Several other mechanisms have been identified as playing major role in the enhanced Raman signal on the ZnO nanorods compared with the other substrates. 

The paper is quite comprehensive and shows interesting results, worth publishing.

Figure 9 regarding the schematics of the multi-effect Raman .enhanced mechanism is unclear and needs better labelling and more clear explanation about these effects that are shown to influence the enhancement of the Raman spectrum. As it stands, the added value of the Figure 9 is not obvious.

Provided these issues are addressed I recommend publication of this paper in Nanomaterials. 

Reviewer 2 Report

The manuscript contains new results, however, some improvement should be done before publication.

1.      The title. I suggest to change slightly the title to “Multi-effect Enhanced Raman Scattering Based on Au/ZnO Nanorods Structures”.

2.      Introduction. The article should be discussed and cited: Fernando, JFSShortell, MPGolberg, DV. Photocatalysis with Pt-Au-ZnO and Au-ZnO Hybrids: Effect of Charge Accumulation and Discharge Properties of Metal Nanoparticles. Langmuir 34 (2018) 7334-7345.

3.      SEM micrographs of four kinds of substrates before Au sputtering should be added for comparison.

4.      Page 4. The sentence “As shown in Figure 1a, when the sputtering time was 20 s, Au particles with extremely small sizes appeared indistinctly on the substrate.” contains wrong information.

5.      SEM micrographs of Au@quartz substrates with different sputtering time should be also added to the manuscript.

6.      Figure 8. The title should be slightly changed to: Figure 8. Absorption spectra of Au nanoparticles on quartz substrates with different sputtering time.

7.      Figure 8. The sputtering time should be added and specified.

8.      The English should be re-checked through whole manuscript very carefully.

The manuscript needs major revision.

Round 2

Reviewer 1 Report

The paper has been improved according to the reviewers suggestion. It is now acceptable for publication.

Reviewer 2 Report

The revised manuscript is suitable for publication.